# Socioeconomic Inequalities and Ethnicity Are Associated with a Positive COVID-19 Test among Cancer Patients in the UK Biobank Cohort

**DOI:** 10.3390/cancers13071514

**Published:** 2021-03-25

**Authors:** Shing Fung Lee, Maja Nikšić, Bernard Rachet, Maria-Jose Sanchez, Miguel Angel Luque-Fernandez

**Affiliations:** 1Department of Clinical Oncology, Tuen Mun Hospital, New Territories West Cluster, Hospital Authority, Hong Kong; leesfm@ha.org.hk; 2Department of Non-Communicable Disease Epidemiology, London School of Hygiene and Tropical Medicine, Keppel Street, London WC1E 7HT, UK; maja.niksic@lshtm.ac.uk (M.N.); bernard.rachet@lshtm.ac.uk (B.R.); 3Inequalities in Cancer Outcomes Network, London School of Hygiene and Tropical Medicine, Keppel Street, London WC1E 7HT, UK; 4Department of Non-Communicable Disease and Cancer Epidemiology, Instituto de Investigacion Biosanitaria de Granada (ibs.GRANADA), University of Granada, 18071 Granada, Spain; mariajose.sanchez.easp@juntadeandalucia.es; 5Cancer Registry of Granada, Andalusian School of Public Health, 18011 Granada, Spain; 6Centro de Investigación Biomédica en Red de Epidemiología y Salud Pública (CIBER of Epidemiology and Public Health, CIBERESP), 28029 Madrid, Spain

**Keywords:** coronavirus disease 2019 (COVID-19), epidemiology, pandemics, cancer, risk factors

## Abstract

**Simple Summary:**

There is limited evidence regarding the influence of socioeconomic factors on COVID-19 transmission, severity and outcomes in the overall population. Furthermore, there is an urgent need to identify and explore the most important socioeconomic risk factors associated with the COVID-19 disease among incident cancer patients, one of the most vulnerable groups of the population. Findings from this study provide invaluable evidence needed for risk classification and stratification among incident cancer patients, based on the information from the first pandemic wave in the UK. We identified the clinical and socio-demographic profile of cancer patients at increased risk of COVID-19 infection. The results from the study added knowledge on impact of the pandemic on the most vulnerable cancer patients in the UK, and can shed light on possible treatment and prevention strategies for COVID-19, including future vaccination prioritisation policy.

**Abstract:**

We explored the role of socioeconomic inequalities in COVID-19 incidence among cancer patients during the first wave of the pandemic. We conducted a case-control study within the UK Biobank cohort linked to the COVID-19 tests results available from 16 March 2020 until 23 August 2020. The main exposure variable was socioeconomic status, assessed using the Townsend Deprivation Index. Among 18,917 participants with an incident malignancy in the UK Biobank cohort, 89 tested positive for COVID-19. The overall COVID-19 incidence was 4.7 cases per 1000 incident cancer patients (95%CI 3.8–5.8). Compared with the least deprived cancer patients, those living in the most deprived areas had an almost three times higher risk of testing positive (RR 2.6, 95%CI 1.1–5.8). Other independent risk factors were ethnic minority background, obesity, unemployment, smoking, and being diagnosed with a haematological cancer for less than five years. A consistent pattern of socioeconomic inequalities in COVID-19 among incident cancer patients in the UK highlights the need to prioritise the cancer patients living in the most deprived areas in vaccination planning. This socio-demographic profiling of vulnerable cancer patients at increased risk of infection can inform prevention strategies and policy improvements for the coming pandemic waves.

## 1. Introduction

The coronavirus disease 2019 (COVID-19), declared as a global pandemic by the World Health Organization (WHO) on 11 March 2020 [1], has introduced a huge strain on communities and health care systems worldwide. With the second or third pandemic wave many countries are finding it difficult to respond to increasing health care needs of people infected with COVID-19, especially if they are also diagnosed with other chronic diseases.

Cancer patients have a higher risk of COVID-19 infection [2], with an estimated 2-fold increased risk of having a positive COVID-19 test, in comparison with the general population [3]. They are also more susceptible to severe COVID-19 infection, followed by higher morbidity and mortality, than people without cancer [2,4,5,6]. The main underlying reason behind this is likely to be their immunosuppressive state, as a result of underlying malignancy or anticancer treatments [7]. Therefore, it is necessary to ensure early detection of coronavirus infection given that cancer patients have a high risk of unfavourable outcomes.

A study using the UK Biobank data found that socio-demographic factors were relevant in determining a risk of COVID-19 infection in the general population, including lower educational attainment and non-white ethnicity [8]. However, the influence of socioeconomic factors on COVID-19 transmission, severity and outcomes among cancer patients are not yet known [9]. We aimed to identify and explore the most important socioeconomic risk factors associated with testing positive for COVID-19 among incident cancer patients in the UK Biobank cohort.

## 2. Materials and Methods

### 2.1. Study Design and Study Participants

We developed a case-cohort study with a modified case-base sampling design consisting in using the complete cohort as the control population [10]. The case-base design involves using an extracted sample from the source population as controls, in order that every person has the same chance of being included as a control, i.e., cases can be included as controls. This type of sampling only works with a previously defined cohort and the inclusion of the complete cohort as control source prevents the use of weights to upweight sampled controls [10]. The case-base sampling provides a valid estimate of the risk ratio (RR), without assuming that the disease is rare in the source population (Appendix A) [10].

Study participants were identified using the UK Biobank—a population-based prospective cohort, which recruited 502,655 volunteers aged 40–69 years, from 2006 to 2010 [11]. Each participant provided longitudinal data on sociodemographic, lifestyle, and behavioural factors, medical history and medications, enabling research into genetic and lifestyle determinants of common diseases of middle and older age [11].

From all the UK Biobank participants we selected those who were diagnosed with incident cancer regardless of the tumour site, defined as cancer cases diagnosed after enrolment to the UK Biobank cohort in contrast to the prevalent cases diagnosed before the enrolment. We excluded non-melanoma skin cancers and other types of non-malignant neoplasms (ICD-10 C.44 and D.00-49). Patients who died before the first available COVID-19 test results (i.e., 16 March 2020) were excluded to allow all participants to have the same probability to be exposed to Severe Acute Respiratory Syndrome Coronavirus 2 (SARS-CoV-2). Dates of death were obtained from death certificates held by the NHS Information Centre (England and Wales) and the NHS Central Register (Scotland).

Data on cancer diagnoses were obtained by UK Biobank through NHS Digital and Public Health England (PHE) for participants in England and Wales, and NHS Central Register and Information Services Division for participants in Scotland [12,13]. Previous evidence linking routine registration of colorectal, lung, and breast cancer with information from the Hospital Episode Statistics showed that the completeness of case ascertainment in English cancer registries exceeds 98% [14]. From the selected sub-cohort of incident cancer, we then identified our cases as all those who tested for SARS-CoV-2 with a positive result based on data from the PHE’s Second Generation Surveillance System. It is a centralised microbiology database covering English clinical diagnostics laboratories that provides national surveillance of legally notifiable infections, bacterial isolations and antimicrobial resistance [15]. Through the individual-level linkage of these two systems, PHE provides a regular and exhaustive feed of new COVID-19 test results for the UK Biobank participants [16]. The majority of samples tested for COVID-19 among UK Biobank participants were from combined nose/throat swabs, while in intensive care units, lower respiratory samples may have been taken [17]. Inpatient tests arise from specimens collected from an acute care provider, an emergency department or an inpatient location [17]. Samples were analysed using a reverse transcriptase polymerase chain reaction (RT-PCR) test for SARS-CoV-2 [17].

### 2.2. Case-Base Constitution and Case-Control Definition

The first COVID-19 test results were released from 16 March 2020 onwards until the 31 August 2020, as part of the first wave of the COVID-19 pandemic. Therefore, the cohort was defined as all the UK Biobank participants diagnosed with an incident cancer and confirmed to be alive on 16 March 2020. Figure 1 shows the study flowchart, including 89 COVID-19 positive cases and 18,917 controls. Cases were defined as all incident cancer patients tested for COVID-19 with a positive result. Controls were defined as the complete cohort of incident cancer cases from the UK Biobank study (i.e., those patients that were not tested for COVID-19 and those patients tested with a positive or negative result). We assume that non-tested controls do not develop the disease.

### 2.3. Sociodemographic Characteristics and Risk Factors at Baseline

The following potential COVID-19 risk factors were identified and referred as time-fixed covariates at baseline: the Townsend Deprivation Index (TDI), age, sex, ethnicity, employment status, smoking status, body mass index (BMI) in kg/m^2^, and type of cancer.

The TDI was the main risk factor, defined as an area-based measure of socio-economic deprivation and a proxy of individuals’ socioeconomic deprivation. TDI scores were calculated using data from the UK 2011 Census. The TDI score combines information on car ownership, household overcrowding, owner occupation, and unemployment aggregated for small areas of residence [18]. Higher TDI equates to a higher level of socioeconomic deprivation. We computed the quintiles (Q) of TDI, and categorised it as Q1 for the least deprived and Q5 for the most deprived group. Age was calculated from dates of birth to baseline assessment. Ethnicity was self-reported and categorised as white or white British, black or black British, Asian or Asian British, and others, including mixed ethnicities and all other, ethnic groups. Smoking status was categorised into never, former, and current smokers. The WHO criteria were applied to classify BMI as underweight (<18.5 kg/m^2^), normal weight (18.5–24.9 kg/m^2^), overweight (25.0–29.9 kg/m^2^), and obese (≥30.0 kg/m^2^) [19]. However, due to data sparsity we recategorised BMI as obese (BMI ≥ 30.0 kg/m^2^) vs. non-obese (BMI < 30.0 kg/m^2^). Employment was categorised as employed or self-employed (baseline category), retired, and unemployed or unpaid. Types of cancers were classified as haematological, non-haematological, melanoma of the skin, and other types of cancer. This categorisation was not based solely on clinical characteristics and largely on the number of positive COVID-19 cases per each distinct cancer type (it had to be >20 diagnosed cases). Finally, we included the duration of time following the cancer diagnosis, classified as being diagnosed with a malignant tumour less than five or more than five years ago.

### 2.4. Statistical Analyses

We described the categorical variables using counts and proportions and the continuous variables using medians and interquartile ranges by case control status. Afterwards, we estimated the risk of testing COVID-19 positive by all the sociodemographic, employment status, BMI and cancer type and time duration after the cancer diagnosis. Due to the use of the complete cohort as main control source for analysis we did not include weights to correct for the sampling design. However, the variance estimates were adjusted (i.e., robust standard errors) to account for the inclusion of the cases as controls in the overall cohort [20]. Therefore, we computed univariate risk ratios and their respective robust 95% confidence intervals (CIs) [20,21].

To assess the association between socio-economic deprivation and the risk of testing positive for COVID-19, we fitted different multivariate regression models, including one variable at a time, to control for confounding (i.e., Models 1–8). The final model was adjusted for age, sex, ethnicity, employment, marital and smoking status, cancer types and years of cancer diagnosis. From each model, we derived the risk ratios (RRs) and 95% CIs using a generalised linear model with the family Poisson and link log and robust error variance estimation [20,21]. From the final adjusted model (Model 8) we derived the adjusted marginal probabilities of testing COVID-19 positive by levels of deprivation (Q5 vs. Q1) stratified by gender and across the levels of BMI.

Given the low proportion of missing data, we first performed a complete-case analysis, adopting the missing completely-at-random assumption. However, to assess the consistency of our results for the socioeconomic deprivation, we developed a multiple imputation by chained equations. We imputed 50 datasets for the variable’s ethnicity, smoking, employment, and marital status. The multiple imputation results were combined using Rubin’s rules [22]. These estimates were presented in Model 9.

Furthermore, in sensitivity analyses we explored different modelling specifications and the assumption that non-tested controls do not develop the COVID-19 (Appendix A). We used Stata v.16 (Stata Corp, College Station, TX, USA) for statistical analysis.

### 2.5. Ethics Approval and Consent to Participate

This UK Biobank study was approved by the North West Multi-Centre Research Ethics Committee (application number 48860); all participants provided written informed consent for data collection, data analysis, and record linkage.

## 3. Results

The characteristics of the test positive (cases) (*N* = 89) and the entire cohort (controls) (*N* = 18,917) are detailed in Table 1. The overall incidence in our study was 4.7 cases per 1000 incident cancer patients (95% CI: 3.8–5.8). Overall, 4090 (21.6%) participants were in the most deprived category, and 3604 (19.1%) in the most affluent group. The median age of the test positive group was 74 years (interquartile range [IQR], 68–78 years) and 52.8% were male. The majority of them were white (91.0%), while 4.5% were black. 74 of 89 test-positive patients (83.1%) had non-haematological cancer, of which malignant neoplasm of the breast, prostate, colorectal, and urogenital were the most common primary tumour sites (Appendix A). We did not find statistically significant differences between the cases and controls regarding the distributions of cancer types, or the duration of time following the cancer diagnosis.

In univariate analysis, black patients had nearly four times higher risk of testing COVID-19 positive than whites (i.e., crude risk ratio [cRR] of 3.77 and 95% confidence interval [CI]: 1.39–10.20). Individuals living in more deprived areas had approximately 4 times higher risk of testing positive for COVID-19 than those individuals from the least deprived areas (i.e., the least deprived quintile of the Townsend index (Q5) vs. the most affluent quintile (Q1) was 3.40; 95% CI: 1.56–7.38) (Table 2). Furthermore, being unemployed or unpaid, current smoker or ex-smoker, having high BMI, or being diagnosed with haematological cancers were strongly associated with an increased risk of testing positive for COVID-19.

In multivariate analysis, we controlled for confounding adding successively one variable at a time in the multivariate regression model (Table 3). Our final complete-case model (Model 8) found a consistent and independent association between being socioeconomically deprived (TDI Q5) and testing positive for COVID-19, with an adjusted risk ratio (aRR) of 2.52 (95% CI 1.00–6.33). Black patients had almost 6 times higher risk of testing positive for COVID-19 in comparison with white patients (i.e., aRR 5.79; 95% CI 1.88–17.85). Furthermore, this risk among unemployed was two times higher than among employed patients (aRR 2.35; 95% CI 1.06–5.20) and increased by 41% for each 5 kg/m^2^ increase in BMI (aRR 1.41; 95% CI 1.20–1.67) (Table 3: Model 8).

Based on the risks estimated from Model 8, we predicted the probabilities of testing positive for COVID-19 for the most and least deprived individuals of the studied cohort, given their respective observed distribution of the covariables included in the model. The probabilities are stratified by sex (Figure 2) and BMI (Figure 3). Overall, we found a clear deprivation gap, but the figures show no evidence of significant differences in COVID-19 positive tests between men and women. Differences between the two sexes were, however, more pronounced among the most deprived group (Figure 2). Figure 3 shows a constant deprivation gap in the probability of testing positive between the most and least deprived cancer patients that widens with increasing levels of BMI and even strengthens among obese cancer patients (i.e., BMI > 30.0 kg/m^2^). Results of the sensitivity analysis are detailed in Appendix A. The effect estimates for being most deprived were consistent in sensitivity analyses based on controls consisting of either test negative or non-tested participants.

## 4. Discussion

We found a consistent and independent association between socioeconomic deprivation and the risk of COVID-19 positive test results among incident cancer cases in the UK Biobank study. Participants with an increased risk of testing positive for COVID-19 during the first wave of the pandemic in the UK had a black ethnic background, lived in the most socioeconomically deprived areas, were obese, unemployed and diagnosed with a haemopoietic cancer, for less than five years.

Our results are consistent with the stark ethnic inequalities evident in the recent estimates of COVID-19 mortality during the first wave of the pandemic by ethnic groups in England and Wales [23,24]. As published by the Office for National Statistics in the UK, the rate of COVID-19 related deaths was 2.7 times higher for black African men then for white men [23]. Thus, we argue that an increased risk of testing COVID-19 positive among black ethnic groups may partially explain the increased mortality risk observed in the UK. Furthermore, the incidence rate in our study (i.e., 4.7 cases per 1000 people) was consistent with the reported figure for the whole UK. The total population in the UK was 66,796,800 (mid-year estimate in 2019 according to the Office for National Statistics of the UK) and the total number of cases as of the 1st August 2020 was 303,942 [25,26], corresponding to an incidence rate was 4.6 per 1000 people. COVID-19 incidence was similar in both populations (general and cancer patients) while external contacts were lower for cancer patients. The is likely to be due to cancer patients’ caution (“shielding”), dramatically reduced delivery of non-COVID-19-related care by the NHS during the first wave of the pandemic, the fear of healthcare interactions during the COVID-19 pandemic or misunderstanding that healthcare services were not available to all but limited to COVID-19 patients only [27,28].

Health surveys in the UK have shown that unemployment, smoking, and obesity are most common among ethnic minorities [24,29]. Stress associated with an unhealthy lifestyle or lack of financial security may result in a pro-inflammatory condition, increasing susceptibility to infections [30,31]. In a study by Shree et al. [32,33], long-term cancer survivors have experienced increased incidence of many infections, have particularly viral and fungal infections among diffuse large B-cell lymphoma and Hodgkin lymphoma survivors. The immune dysfunction can be related to long-lasting changes in the functioning of the immune system due to cancer diagnosis and past treatments. The increased risk for COVID-19 infection found among the most socioeconomically deprived incident cancer patients could point to a potential role for proinflammatory and metabolic condition in combination with cancer-related immune dysfunction leading to increased vulnerability to infections.

Socioeconomic status is a multifaceted construct, and therefore, not only determined by the individuals’ income. We argue that the disparities in testing positive for COVID-19 we found during the first wave of the pandemic reflect the health, environmental, and occupational effects of socioeconomic inequalities. These disparities may be partially explained by the higher likelihood of ethnic minorities to work in lower paid occupations that demand proximity to other people (e.g., delivery staff, security or cleaning services, social or health care assistants) or work colleagues (e.g., construction industry), offering less flexibility to work from home, or living in cramped residential settings that preclude social distancing (i.e., several generations living together) [34,35,36,37,38,39].

Our study highlights the necessity of reporting data on socioeconomic determinants of COVID-19 disease, especially information about ethnicity to identify high-risk populations and develop equitable public health prevention measures, guidelines, and interventions. A multi-sectoral approach would reduce disparities by considering the vulnerabilities at a social, educational, economic, and provision-of-care levels. Furthermore, findings from our study may help to prioritise future COVID-19 preventive strategies, especially vaccination priority, among cancer patients undergoing active treatments However, more research is needed regarding comparative risk assessment for COVID-19 infection among cancer and the overall population and the cancer specific survival after COVID-19 infection.

UK Biobank has been meticulously tracking the health status of its participants for over a decade. Many participants are now in the age range likely to be most vulnerable to COVID-19 infection as well as at higher risk of being diagnosed with cancer. Using the case-cohort approach we derived risk estimates for positive COVID-19 test results based on a valid and consistent methodological approach [40,41]. However, there are possible issues around generalisability of the UK Biobank data, which was previously criticised [42]. Although the UK Biobank might not be completely representative of the UK population and therefore the results may not be entirely generalizable, or suitable for identifying disease prevalence or incidence rates, its large size and heterogeneity of exposure measures provide valid scientific inferences of associations between exposures and health conditions that are generalizable to other populations [42]. Furthermore, we were unable to perform cancer specific analysis due to small number of confirmed COVID-19 cases by cancer sites. However, we were able to classify malignancies in clinically relevant categories, i.e., haematological cancers, non-haematological cancers, skin melanoma, and other types of cancers. The classification was not based on clinical characteristics; rather it was based on the number of positive COVID-19 cases per each distinct cancer type (it had to be >20 diagnosed cases).Finally, we assume that controls that were not tested for COVID-19 do not develop the disease which may have induced a selection bias because some of the cancer patients might be asymptomatic [43]. We argue that given their cancer status the majority of control patients developing symptoms compatible with COVID-19 they would have been tested, also we have performed sensitivity analyses demonstrating consistency of results even if non-tested patients were excluded from analysis. Furthermore, in our sensitivity analysis, the results were consistent when restricting the controls to only non-tested patients excluding those testing negative on COVID-19 (comparing test positive versus non-tested patients).

## 5. Conclusions

To the best of our knowledge, this study is the first to highlight a consistent and independent association between socioeconomic deprivation and the risk of COVID-19 infection among incident cancer patients in the UK. Policy and practice improvements based on high ethical principles are needed to address the broad disparity and risk stratification among the most vulnerable cancer patients at risk for COVID-19. It is essential to develop effective preventive measures targeting cancer patients at highest risk, such as an urgent vaccination of the underprivileged cancer patients, to confront the future potential pandemic waves.

## Figures and Tables

**Figure 1 cancers-13-01514-f001:**
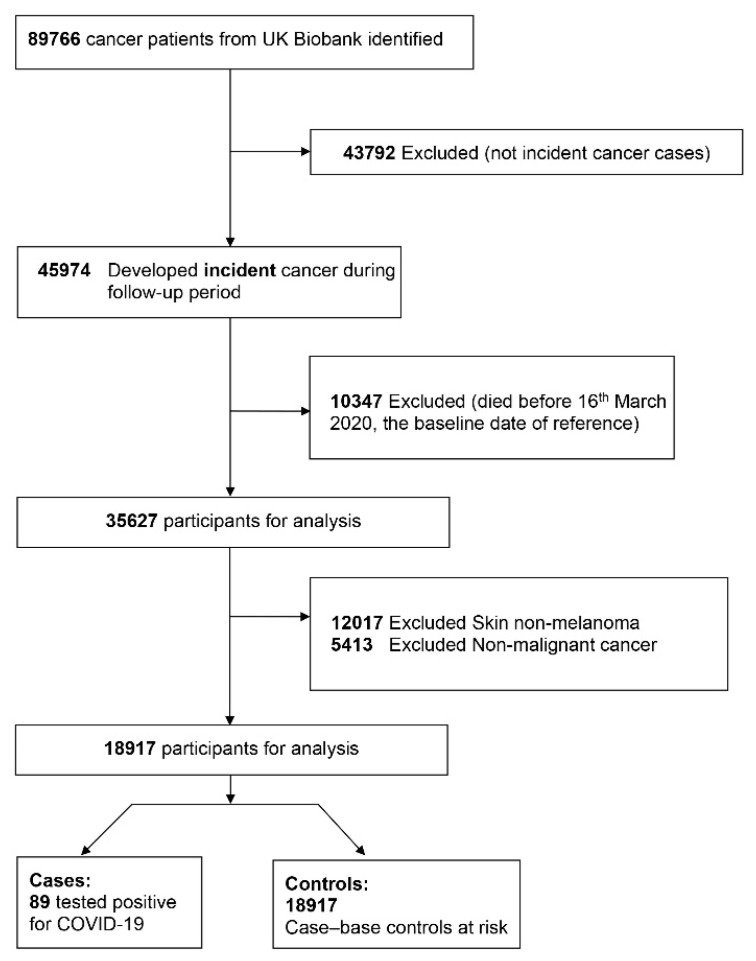
Case-cohort UK Biobank Study flowchart.

**Figure 2 cancers-13-01514-f002:**
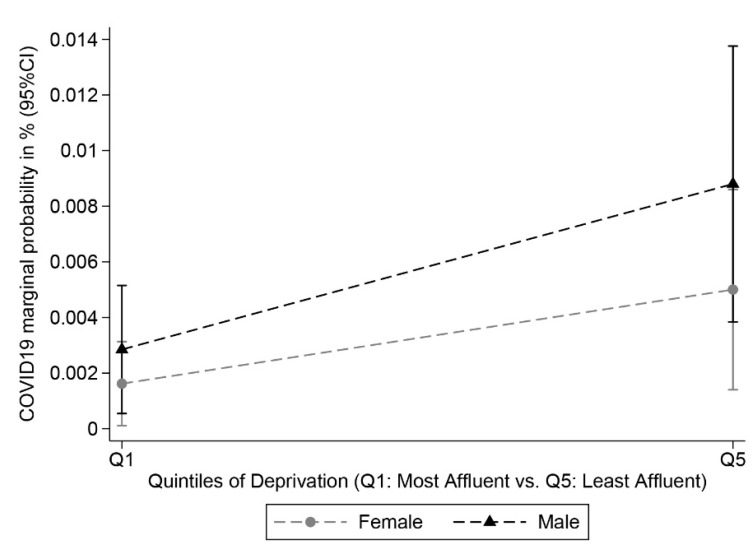
Probabilities of COVID-19 Testing Positive among UK Biobank Incident Cancer Patients by deprivation (most affluent vs. least affluent) and sex (*N* = 18,917, 89 cases).

**Figure 3 cancers-13-01514-f003:**
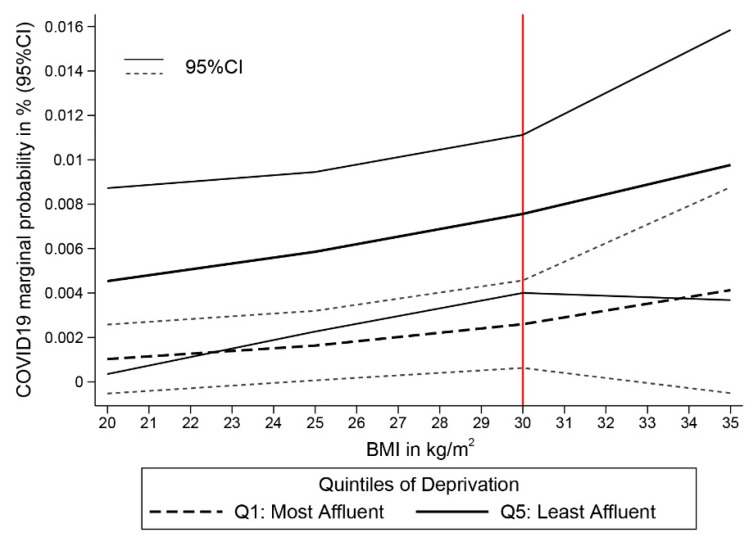
Probabilities of COVID-19 Testing Positive among UK Biobank Incident Cancer Patients by deprivation (most affluent vs. least affluent) and BMI in kg/m^2^ (*N* = 18,917, 89 cases).

**Table 1 cancers-13-01514-t001:** Case-Cohort Control Study Socioeconomic Characteristics, Smoking Status BMI, and Cancer Type Among UK Biobank Incident Cancer Patients at Baseline on 16 March 2020 (*N* = 18,917, 89 cases).

Characteristics	Tested Positive (Cases) *N* (%)	Entire Cohort (Controls) *N* (%)	*p*-Values
**Age, years**			0.177 *
Median (interquartile range)	74 (68–78)	73 (67–77)	
**Quintiles of Deprivation**			0.013
1st quintile (most affluent)	8 (9.0)	3604 (19.1)	
2nd quintile	16 (18.0)	3682 (19.5)	
3rd quintile	18 (20.2)	3659 (19.3)	
4th quintile	15 (16.9)	3863 (20.4)	
5th quintile (most deprived)	31 (34.8)	4090 (21.6)	
**Sex**			0.518
Female	42 (47.2)	9577 50.6)	
Male	47 (52.8)	9340 (49.4)	
**Ethnicity**			0.031
White	81 (91.0)	18,075 (95.6)	
Asian	2 (2.3)	284 (1.5)	
Black	4 (4.5)	234 (1.2)	
Others	2 (2.3)	220 (1.2)	
**Marital status**			0.726
With a partner	68 (76.4)	14,048 (74.3)	
Without a partner	5 (5.6)	1215 (6.4)	
**Employment**			0.031
Employed or self-employed	33 (37.1)	9240 (48.8)	
Retired	42 (47.2)	8197 (43.3)	
Unemployed/unpaid	11 (12.4)	1283 (6.8)	
**Smoking status**			0.022
Current smoker	12 (13.5)	1767 (9.3)	
Ex-smoker	43 (48.3)	7251 (38.3)	
Non-smoker	33 (37.1)	9772 (51.7)	
**BMI (kg/m^2^)**			0.001 *
Median (interquartile range)	28.6 (26.1–31.9)	27.0 (24.5–30.1)	
**Categorical BMI**			0.026
<18.5	0 (0.0)	64 (0.3)	
18.5–24.9	16 (18.0)	5557 (29.4)	
25.0–29.9	38 (42.7)	8412 (44.5)	
≥30	33 (37.1)	4795 (25.4)	
**Malignancy types**			0.069
Haematological cancers	12 (0.8)	1518 (99.2)	
Non-haematological cancers	74 (0.5)	15,975 (99.5)	
Skin melanoma and others	3 (0.2)	1424 (99.8)	
**Years of cancer diagnosis**			0.052
Within 5 years of diagnosis	21 (23.6)	3027 (16.0)	
Beyond 5 years of diagnosis	68 (76.4)	15,883 (84.0)	
**Total numbers**	89 (0.5)	18,917 (100.0)	
**Missing values**			
Quintiles of deprivation	1 (1.1)	19 (0.1)	
Ethnicity	0 (0.0)	104 (0.6)	
Marital status	16 (18.0)	3654 (19.3)	
Employment	3 (3.4)	197 (1.0)	
Smoking	1 (1.1)	127 (0.7)	
BMI	2 (2.2)	89 (0.5)	
Years of cancer diagnosis	0 (0.0)	7 (0.0)	

* Kruskal-Wallis rank test. BMI: Body Mass Index in kg/m^2^.

**Table 2 cancers-13-01514-t002:** Case-Cohort Control Study Unadjusted Risk Ratios of Testing COVID-19 Positive among UK Biobank Incident Cancer Patients by Deprivation, Age, Sex, Ethnicity, Employment Status, BMI and Cancer Types and Duration (*N* = 18,917 and 89 cases).

Characteristics	Crude RR (95% CI)	*p*-Value
**Age**		
per ten-year increase	1.17 (0.83–1.64)	0.353
**Sex**		
Male vs. Female	1.15 (0.76–1.74)	0.518
**Ethnicity**		
Asian vs. White	1.57 (0.39–6.34)	0.529
Black vs. White	3.77 (1.39–10.20)	0.009
Others vs. White	2.02 (0.50–8.16)	0.324
**Marital status**		
With a partner vs. Without partner	1.18 (0.47–2.91)	0.727
**Quintiles of Townsend deprivation index**		
2nd quintile vs. 1st quintile	1.95 (0.84–4.56)	0.121
3rd quintile vs. 1st quintile	2.21 (0.96–5.08)	0.062
4th quintile vs. 1st quintile	1.75 (0.74–4.11)	0.202
5th quintile vs. 1st quintile	3.40 (1.56–7.38)	0.002
**Employment status**		
Retired vs. employed	1.43 (0.91–2.26)	0.122
Unemployed/unpaid vs. employed	2.39 (1.21–4.71)	0.012
**Smoking status**		
Current smoker vs. non-smoker	2.00 (1.04–3.87)	0.039
Ex-smoker vs. non-smoker	1.75 (1.11–2.75)	0.015
**BMI**		
(per five kg/m^2^ increase)	1.30 (1.08–1.56)	0.006
**Categorical BMI in kg/m^2^**		
Obesity (≥30) vs. Normal/overweight (<30)	1.78 (1.16–2.75)	0.009
**Malignancy types**		
Hematological cancer vs. Skin melanoma/others	3.73 (1.05–13.19)	0.041
Non-hematological cancer vs. Skin melanoma/others	2.19 (0.69–6.95)	0.182
**Years of cancer diagnosis**		
Within 5 years vs. Beyond 5 years of diagnosis	1.62 (0.99–2.63)	0.054

Abbreviations: BMI, Body Mass Index; RR, Risk Ratio.

**Table 3 cancers-13-01514-t003:** Case-Cohort Control Study Adjusted Risk Ratios of Testing COVID-19 Positive among UK Biobank Incident Cancer Patients by Deprivation, Age, Sex, Ethnicity, Employment Status, BMI and Cancer Type (*N* = 18,917).

Variables	Model 1 aRR (95% CI)	Model 2 aRR (95% CI)	Model 3 aRR (95% CI)	Model 4 aRR (95% CI)	Model 5 aRR (95% CI)	Model 6 aRR (95% CI)	Model 7 aRR (95% CI)	* Model 8 aRR (95% CI)	Model 9 aRR (95% CI)
**Townsend Deprivation Index**									
2nd vs. 1st	1.96 (0.84–4.58)	1.96 (0.84–4.58)	1.84 (0.78–4.34)	1.81 (0.77–4.28)	1.79 (0.76–4.23)	1.80 (0.76–4.25)	1.79 (0.76–4.24)	2.09 (0.85–5.13)	1.90 (0.82–4.46)
3rd vs. 1st	2.22 (0.97–5.11)	2.21 (0.96–5.09)	2.21 (0.96–5.08)	2.18 (0.95–5.01)	2.01 (0.86–4.68)	2.02 (0.87–4.69)	2.00 (0.86–4.65)	2.02 (0.81–5.04)	1.98 (0.85–4.61)
4th vs. 1st	1.78 (0.75–4.19)	1.74 (0.74–4.07)	1.74 (0.74–4.07)	1.56 (0.65–3.72)	1.49 (0.63–3.54)	1.49 (0.63–3.53)	1.49 (0.63–3.54)	1.78 (0.72–4.42)	1.60 (0.68–3.76)
5th vs. 1st	3.46 (1.59–7.53)	3.20 (1.45–7.06)	2.86 (1.27–6.41)	2.69 (1.21–6.00)	2.41 (1.05–5.55)	2.38 (1.04–5.48)	2.37 (1.03–5.44)	2.52 (1.00–6.33)	2.57 (1.13–5.85)
**Sex**									
male vs. female	1.13 (0.74–1.74)	1.11 (0.72–1.72)	1.16 (0.74–1.83)	1.12 (0.72–1.75)	1.16 (0.73–1.85)	1.16 (0.73–1.85)	1.16 (0.73–1.83)	1.20 (0.71–2.02)	1.08 (0.69–1.70)
**Age**									
per ten-year increase	1.17 (0.83–1.73)	1.21 (0.86–1.70)	1.15 (0.73–1.81)	1.12 (0.70–1.80)	1.10 (0.68–1.78)	1.08 (0.67–1.76)	1.10 (0.68–1.78)	1.10 (0.66–1.84)	1.12 (0.70–1.80)
**Ethnicity**									
Asian vs. White		1.49 (0.36–6.04)	0.74 (0.10–5.41)	0.91 (0.12–6.61)	0.97 (0.13–7.39)	0.93 (0.12–7.09)	0.94 (0.13–6.89)	1.16 (0.14–9.31)	1.64 (0.39–6.89)
Black vs. White		2.92 (0.99–8.67)	2.97 (1.00–8.85)	3.43 (1.21–9.77)	3.52 (1.19–10.45)	3.39 (1.14–10.07)	3.41 (1.15–10.11)	5.79 (1.88–17.85)	3.44 (1.13–10.46)
Others vs. White		1.86 (0.47–7.30)	1.80 (0.46–7.05)	1.89 (0.46–7.82)	2.00 (0.51–7.83)	1.94 (0.50–7.58)	1.91 (0.49–7.46)	2.83 (0.73–10.91)	1.80 (0.46–7.06)
**Employment status**									
Retired vs. employed			1.29 (0.74–2.24)	1.32 (0.76–2.31)	1.28 (0.72–2.26)	1.28 (0.72–2.27)	1.28 (0.72–2.26)	1.39 (0.79–2.46)	1.22 (0.69–2.14)
Unemployed/unpaid vs. employed			2.09 (1.03–4.24)	2.11 (1.04–4.29)	2.01 (0.99–4.09)	2.00 (0.98–4.07)	1.99 (0.98–4.04)	2.35 (1.06–5.20)	2.07 (1.02–4.20)
**Smoking status**									
Current smoker vs. non-smoker				1.44 (0.71–2.96)	1.51 (0.74–3.06)	1.49 (0.73–3.02)	1.49 (0.73–3.04)	1.30 (0.54–3.16)	1.77 (0.91–3.44)
Ex-smoker vs. non-smoker				1.69 (1.07–2.68)	1.57 (0.99–2.50)	1.56 (0.98–2.49)	1.56 (0.98–2.49)	1.53 (0.93–2.52)	1.56 (0.98–2.47)
**BMI**									
Per 5 kg/m^2^ increase					1.28 (1.08–1.52)	1.29 (1.09–1.52)	1.28 (1.08–1.51)	1.41 (1.20–1.67)	1.27 (1.08–1.50)
**Malignancy type**									
Haematological vs. skin melanoma and ** others						4.21 (0.92–19.33)	4.18 (0.91–19.18)	3.94 (0.84–18.38)	4.30 (0.94–19.70)
Non-haematological vs. skin melanoma and ** others						2.74 (0.67–11.22)	2.74 (0.67–11.19)	2.31 (0.56–9.53)	2.80 (0.69–11.46)
**Years of cancer diagnosis**									
Within 5 years vs. beyond 5 years of diagnosis							1.55 (0.93–2.58)	1.44 (0.81–2.56)	1.67 (1.02–2.73)
**Marital status**									
With a partner vs. without a partner								2.36 (0.66–8.35)	1.36 (0.52–3.57)

Abbreviations: aRR, adjusted risk ratio; BMI, body mass index; M1: adjusted for Townsend Deprivation Index, age, and sex; M2: M1 + Ethnicity; M3: M2 + Employment status; M4: M3 + Smoking status; M5: M4 + BMI in kg/m^2^; M6: M5 + Malignancy type; M7: M6 + Years of cancer diagnosis; M8: M7 + Marital status; M9: Multiple imputation. * Model 8 is the final model. ** others: all other cancers excluding non-melanoma skin cancers and other types of non-malignant neoplasms (ICD-10 C.44 and D.00-49).

## Data Availability

This research has been conducted using the UK Biobank Resource under application number 48860 granting access to the corresponding UK Biobank biomarkers, and phenotype data. De-identified participant phenotype and genetic data from the UK Biobank data are available via (www.ukbiobank.ac.uk (accessed on 6 September 2020)) on application. Proposals should be directed to the UK Biobank to gain access, data requestors will need to sign a data access agreement.

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
