# Peer review of "Socioeconomic Inequalities and Ethnicity Are Associated with a Positive COVID-19 Test among Cancer Patients in the UK Biobank Cohort"

_cancers, 2021, doi:10.3390/cancers13071514_

Round 1

Reviewer 1 Report

In this article, the authors explored the role of socioeconomic inequalities in COVID-19 incidence among cancer patients. They start out by calling it a nested case-control study within the UK Biobank cohort, from 16 March, 2020 to 23 August, 2020. The main exposure variable was socioeconomic status assessed by the Townsend Deprivation Index. Starting from over 89,000 cases, they narrowed down to 18,917, of which 89 tested positive for COVID-19. The overall COVID-19 incidence was 4.7 cases per 1,000 incident cancer patients (95%CI 3.8–5.8). Compared with the least deprived cancer patients, those living in the most deprived areas had an almost 3 times higher risk of testing positive (RR 2.6, 95%CI 1.1–5.8). Other independent risk factors were ethnic minority background, obesity, unemployment, smoking, and being diagnosed with a haematological cancer for less than 5 years. The authors conclude by identifying a consistent pattern of socioeconomic inequalities in COVID-19 among incident cancer patients in the UK, and a need to prioritize the cancer patients living in the most deprived areas in vaccination planning.

The information generated from this study is now well known. Social and economic inequalities have only become more obvious during this pandemic. The higher incidence rate and higher case fatality rate has been linked to a more deprived status and presence of co morbidities. Despite that, the information is important as it pertains specifically to patients with cancer.

Some weaknesses that need to be addressed

This study does not fit the pattern of a nested case control study. They did not pick controls based on cases. They simply included all relevant patients, after exclusion.

They don’t correct for multiple analysis in the same data.

They present the 9 models in table 3, but only mention 1-8 in text, what is the significant of model 9

They should discuss if risk of COVID is higher or lower or similar to general population. As per current stats, England has 4.17 million cases in a population of 67 million. They can find numbers from Aug 2020, and discuss.

Overall, it is a study with findings worth publishing, but after some changes.

Reviewer 2 Report

I think one of the goals is to make sure the right people get the vaccines first and the argument needs to be a strong as possible. The current article may be lacking key information, which should be available to the authors. I think the findings are interesting and topical - identifying black, socioeconomically deprived, obese, unemployed, and diagnosed with a haemopoietic cancer as associated with a COVID + test.

  1. Not sure all the findings are cancer specific (or should be cancer specific). Would it be more informative to indicate TDI as a predictor in the total database also.
  2. ‘Simple summary’ to be toned down from hyperbole, as it is not on the same level as the introduction.
  3. Major limitation is the COVID incidence rate. The methods should outline the power to look at that many variables given the likely incidence.
  4. Evidence to whether the cancer population is, in itself, vulnerable and in need of the first vaccines? Is the incidence of COVID in the cancer population comparable to the non-cancer cohort? In patients with COVID estimate of median survival (Any difference to the non-cancer population)? Were the identified risk factors consistent in the non-cancer population?
  5. Second and third wave information – less bias to location of clusters, and more cases?
  6. Is there a way to do the analyses stratified by postcode/council area – i.e. try to adjust for locational clusters? i.e. you can only get COVID if it is in your area.
  7. Time bias in follow up after first covid case. Need for time to event analysis?
